# Prevalence of refractive errors and risk factors for myopia among schoolchildren of Almaty, Kazakhstan: A cross-sectional study

Ainagul Mukazhanova[1,2], Neilya Aldasheva[1], Juldyz Iskakbayeva[1], Raushan Bakhytbek[1], Aliya Ualiyeva[2], Kaini Baigonova[2], Damet Ongarbaeva[2], Denis Vinnikov[2,3]*

1 Kazakh Eye Research Institute, Almaty, Kazakhstan, 2 Faculty of Medicine and Health Care, Al-Farabi Kazakh National University, Almaty, Kazakhstan, 3 Peoples' Friendship University of Russia (RUDN University), Moscow, Russian Federation

* denisvinnikov@mail.ru

**Data Availability Statement:** All relevant data are within the paper.

## Abstract

### Introduction

Very little is known about the prevalence of refractive errors among children in Kazakhstan. The aim of this study was to investigate the prevalence of refractive errors and risk factors of myopia among schoolchildren in Almaty, Kazakhstan.

### Methods

In the cross-sectional study of 2293 secondary school students (age 6–16), we examined cycloplegic autorefraction and offered a questionnaire in three age groups: 1st grade (N = 769), 5th grade (N = 768) and 9th grade (N = 756). The questionnaire covered main risk factors such as parental myopia, screen time, time outdoors, sports activities, near work, gender, grade, and school shift. Adjusted logistic regression analysis was applied to test the association of risk factors with myopia.

### Results

The mean spherical equivalent (SER) was -0.54 ± 1.51 diopters (D). The overall prevalence of refractive errors was 31.6% (95% confidence interval (CI) 29.7; 33.5); myopia 28.3% (95% CI 26.5; 30.1); hyperopia 3.4% (95% CI 2.7–4.1) and astigmatism 2.8% (95% CI 2.1; 3.5). In the multivariate adjusted regression analysis, higher class level (5th grade (odds ratio (OR) 1.78; 95% CI 1.26; 2.52) and 9th grade (OR 3.34; 95% CI 2.31; 4.82)) were associated with myopia, whereas outdoors activity more than 2 hours a day (OR 0.64; 95% CI 0.46; 0.89) and sports (OR 0.70; 95% CI 0.52; 0.93) were associated with a lower incidence of myopia.

### Conclusions

Myopia is a leading refractive error in schoolchildren in Almaty, Kazakhstan. Myopia prevention measures, including more time outdoors, should guide public health interventions in this population.

**Funding:** The authors received no specific funding for this work.

**Competing interests:** The authors have declared that no competing interests exist.

## Introduction

Uncorrected refractive errors are the leading cause of moderate to severe visual impairment worldwide and the second most common cause of avoidable blindness [1, 2]. Studies on the prevalence of different types of refractive errors among children in different parts of the world are inconsistent and affected by age, gender, geography, and ethnicity [3–8]. Childhood myopia takes the lead in some countries of Southeast Asia with prevalence reaching 80% among adolescents, whereas hyperopia in children may be most prevalent in the Americas [9, 10]. At present, myopia, in particular childhood myopia, is a major public health issue, which in recent years has grown into an epidemic [4, 11, 12]. Many parts of the world have faced a dramatic increase in the number of people with myopia, especially among children and adolescents in Eastern Asian countries, in recent decades [10, 13–18].

The greatest burden of refractive error is myopia, with high prevalence rates in school-age children and adolescents, even greater in those with higher attained education [13, 19, 20]. According to Holden et al., half the world population (49.8%) will be myopic by the year 2050 and about 9.8% of people will have high myopia [11]. Childhood myopia, especially its early manifestation, increases the risk of complications, such as amblyopia, cataract, glaucoma, retinal detachment and myopic macular degeneration [21–23].

The studies of refractive errors from the countries of the former Soviet Union, especially those situated in Central Asia, are not very abundant, whereas the data from local studies are not available for the international audience. Sporadic reports from the countries of the former Soviet Union show a great variability in myopia prevalence. For example, the prevalence of myopia among urban children in Armenia was 23.3% in 2017 [24], but much greater in urban adolescents in Azerbaijan (34.7%) [25]. The prevalence of myopia in Russia also varies widely from 5.1% to 50.7%, depending on the region and age of the subjects [26–28]. Inconsistent data from the countries across the former Soviet Union may result from varying geographical and climatic conditions in addition to differences in ethnic and cultural composition of the peoples living in these countries [29, 30].

In the last 20 years, a number of studies reported the prevalence of visual impairments in school-age children in Kazakhstan [31–35]. In 2001, a pediatric examination in Almaty showed that 7.4% school-age children had any visual impairments, whereas the prevalence of refractive errors was not studied [34]. A cross-sectional study in the 2004 in Almaty, Kazakhstan revealed a 21% prevalence of refractive errors in schoolchildren, whereas myopia was confirmed in 14%, hyperopia in 3%, astigmatism in 1% and accommodation disorders in 3% of the studied population [31]. Another study (2010) reported that myopic refraction was found in 12% of rural school children and 22% among their urban counterparts [35]. Recently, there were only a few studies of the prevalence and structure of refractive errors among schoolchildren in Kazakhstan.

Therefore, the aim of this study was to ascertain the prevalence of refractive errors and predictors of myopia among school-age children and adolescents living in a metropolitan area of Almaty, Kazakhstan.

## Materials and methods

### Study population and sampling

It was a school-based cross-sectional study conducted from September to December 2019 in Almaty (Kazakhstan). From 1929 to 1997 Almaty was the capital of Kazakhstan. At present, Almaty remains the largest city in Kazakhstan with an area of 682 square kilometers and with a population of about 1.85 million inhabitants (about 10% of the country's total population). In 2018/2019, there were about 470,000 (25.3%) children 17 years old and younger living in Almaty [36, 37].

The educational system of Kazakhstan includes pre-school (pre-primary) education (kindergartens or nursery schools), secondary education, comprising schools for children 6 to 17 years old and higher education which includes universities, institutes, academies, etc. Public schools of Kazakhstan have a double-shift system, in which two separate groups of pupils attend schools during a school day. The first shift is in class from early morning until mid-day, whereas the second shift usually attends from mid-day to late afternoon.

This cross-sectional school-based study was performed according to the protocol of Refractive Error Study in Children (RESC). This protocol was designed to standardize the methodology used to obtain prevalence data on childhood refractive errors [38]. The sample size was calculated as $N = \frac{Z^2 * (1-P) * P}{(B*P)^2}$, where P is the anticipated prevalence of myopia, B is the desired error bound.

Prompted by previous research, the anticipated refractive error prevalence was 20% with a 15% error rate and a 95% confidence interval [35]. With a simple random sampling, 683 children were required for each class-level stratum. Assuming 5% non-participation, the required sample size increased to 751. The non-response rate was assumed to be 10%. Therefore, the calculated sample size was 751 for each of the three class-level strata, which corresponds to a total study sample of 2260 children. Considering that the average size of a class in the sampling frame was 36 (29–43) children, 63 randomly selected classes would be needed in total from three strata, 21 classes from each stratum.

This study was conducted between September to December 2019. We studied three age groups, including 1st, 5th and 9th grades from eight schools, which were randomly selected out of 201 Almaty schools. Sampling was performed maintaining the proportionality between the number of schools and 1st, 5th and 9th grades students. The number of studied classes in each school depended on the total number of students and varied from 2 to 4 grades of each link (on average, from 6 to 12 grades in each school). The study included students from two different types of secondary schools, such as general secondary schools and gymnasiums; however, all schools were state-owned. Both the general education school and the gymnasium were public secondary schools. Gymnasium was an advanced secondary education school with in-depth study of selected classes. These two school types differed in the in core academic hours and the overall structure of the academic workload. All children from the selected schools (N = 2442) were invited to participate in the study, but only 2293 (response rate 93.3%) accepted the invitation and took part in the study. Those children who had a history of any ocular surgery (including intraocular, refractive, trauma and strabismus surgeries), any inflammatory and infectious eye diseases, keratoconus, heterotropia, congenital cataract and pterygium were not included into the study. Students under orthokeratology treatment were excluded because their uncorrected visual acuity and uncorrected refractive error could not be obtained.

## Questionnaire

We also asked children's parents to complete a questionnaire, which included demographic information, history of parental myopia and participants' behavioral factors. Demographic information included age, gender and school grades. Behavioral factors included the average duration of daily gadgets use (computer, mobile phone, tab, games, etc.), near-work (extracurricular activities), and the outdoor activity estimated in hours per day. We used two questions to verify near work: (1) "How many hours does the child spend on the near work (reading, drawing, handicraft, homework, etc.) daily (time spent at school is not considered)?"; (2) "How many hours does the child spend with gadgets (computers, mobile phones, tabs, games, etc.) daily? In addition, outdoor and sport activity were assessed with two questions: (1) "How many hours does the child spend in the outdoor activities daily; (2) "Does your child attend

sports club or section?". Finally, parental myopia was ascertained with a question "Does any parent have myopia?"' with three response options (no; one of the parents; both parents).

## Examination protocol

Clinical examinations were conducted at temporary stations in the school premises. One team examined all children. The clinical team comprised one ophthalmologist and two ophthalmic assistants. The team was trained on the procedure for research in accordance with the RESC recommendations. All children underwent the following standard procedures: distance visual acuity testing, noncycloplegic autorefraction, slit-lamp examination followed by cycloplegia and cycloplegic autorefraction and ophthalmoscopy. The vision examinations and autorefractometry were performed by professional licensed ophthalmic nurses (ophthalmic assistants) in a well-lighted classroom. Visual acuity was measured at a distance of 5 meters, using a retro-illuminated Logarithm of the minimum angle of resolution (LogMAR) chart with Landolt C broken rings and was recorded as the smallest line read with one or no errors. Vision was recorded in decimal notation.

We measured cycloplegic refraction 30 minutes after the instillation of a drop of cyclopentolate 1% twice. Cycloplegia was considered complete if the pupils were dilated 6 mm or greater and the pupillary light reflex was absent. The pupil size was determined roughly with the help of a regular stationery ruler. If needed, a third drop was instilled. Children were asked to keep their eyes closed, for the duration of cycloplegia. We used RC-5000 (Tomey, Tokyo, Japan) for autorefraction with an average of three measurements. The autorefractor was calibrated at the beginning of each working day. Examination of the lens, vitreous, and fundus was performed by an ophthalmologist with a slit lamp and indirect ophthalmoscope.

## Definitions

Refractive errors are determined by the spherical equivalent refraction (SER) calculated as sphere plus the half of the cylindrical error. In accordance with international recommendations, myopia was defined as myopic refractive error when SER of $\leq$-0.50 D [39]. Myopia was further divided into four refractive error groups: low myopia (-3.0 D $\leq$ SER $\leq$-0.50 D), moderate myopia (-6.0 D $\leq$ SER $<$-3.0 D D) and high myopia (SER $<$-6.0 D) [39]. Refractive errors of $\geq$ +1.00 D were classified as hyperopia [40]. Thus, emmetropia was defined as SER in the range -0.50 D $<$ SER $<$ + 1.00 D. Clinically significant astigmatism (CSA) was defined as a cylindrical error as $>$ 1.0 D, regardless of sign [40]. Because the correlation of spherical equivalent and visual acuity between the eyes was high (Pearson correlation coefficient 0.91 (95% CI 0.90; 0.92; p<0.001)), the spherical equivalent from the worse eye was used for analysis, similar to several previous cross-section studies in children [6, 7, 41, 42]. The worse eye was defined as the eye with the greater absolute value of the SER.

## Ethical issues

This study was conducted in accordance with the Declaration of Helsinki, and all procedures involving human subjects were approved by the Ethics committee of the Kazakh Eye Research Institute (No. 4 dated September 23, 2019) and the Ethics Committee of Al-Farabi Kazakh National University (No. 27–2019 dated May 21, 2019).

## Statistical analysis

We tested normality of all data and reported means with the standard deviation or medians with the corresponding interquartile range (IQR), prompted by the data normality. The main

outcome was the prevalence of refractive errors in different groups, which we reported as % with their 95% confidence interval (CI). Between-group comparisons of binary variables were tested using χ2 test from contingency tables, whereas continuous variables were tested using t-test or Mann-Whitney U-test depending on the distribution. We used crude and multiple regression modelling to test the association of refraction errors with selected predictors, identified in the crude analyses. Because some categorical predictors of myopia could have been collinear, we used correlation matrix to test for correlations and found no significant correlations between them. The final adjusted model accounted for seven predictors, including gender, grade, parental myopia, near-work, time spent outdoors, full day of schooling and regular sports activities. The effect in both crude and adjusted analysis was quantified with the odds ratio (OR) with its 95% CI. The cut-off value of a probability of no effect was set to 0.05. All statistical analyses were performed using NCSS (NCSS, Utah, USA).

## Results

In the current cross-sectional study of 2293 analyzed subjects, 50.6% (n = 1161) were boys and 49.4% (n = 1132) were girls. The non-response rate in this study was 6.1%. The mean age of participants was 11.2 ± 3.6 (range 6–16) years. There were no significant differences in gender between groups. Gender, grade and school type distribution is shown in Table 1.

Table 1 summarizes the prevalence of refractive errors among all examined children. The mean spherical equivalent in the total sample was -0.54±1.51 D. There were no significant differences in refractive errors between boys and girls. The prevalence of refractive errors among all children was 31.6% (95% CI 29.7; 33.5) and increased with age and grade, from 22.2% (95% CI 19.30; 25.1) in 1st grade to 43.3% (95% CI 39.8; 46.8) in 9th grade. There were no differences in the prevalence of refractive errors between two types of school.

The overall prevalence of myopia was 28.3% (95% CI 26.5; 30.1). Of all participants with myopia, 79.2% had low myopia, 16.4% had moderate myopia, and 4.5% had high myopia. High myopia was more frequent among students of gymnasiums than among general education schools (6.5% vs 2.7%; p = 0.014), while low myopia was more often observed in students of general education schools. Prevalence of hyperopia was 3.4%. Low hyperopia was found in 89.6% and moderate hyperopia was detected in 10.4% of all hyperopic participants. As demonstrated in Table 2, there were no significant differences in the prevalence of myopia and hyperopia among boys and girls.

**Table 1. The characteristics of sample and prevalence of refractive errors (N = 2293).**

| Demographic variable | N (%) | Mean SER ± SD, D | Emmetropia, n (%) | Refractive error, n (%) | p |
|---|---|---|---|---|---|
| Gender | | | | | |
| Male | 1161 (50.6) | -0.52±1.54 | 802 (69.1) | 359 (30.9) | 0.468 |
| Female | 1132 (49.4) | -0.56±1.48 | 766 (67.7) | 366 (32.3) | |
| Grade | | | | | |
| 1st | 769 (33.5) | -0.19±1.07 | 598 (77.8) | 171 (22.2) | <0.001 |
| 5th | 768 (33.5) | -0.48±1.43 | 541 (70.4) | 227 (29.6) | |
| 9th | 756 (33.0) | -0.95±1.84 | 429 (56.7) | 327 (43.3) | |
| School type | | | | | |
| Gymnasiums | 1062 (46.3) | -0.63±1.61 | 722 (68.0) | 340 (32.0) | 0.705 |
| General education schools | 1231 (53.7) | -0.45±1.42 | 846 (68.7) | 385 (31.3) | |
| Total | | -0.54±1.51 | 1568 (68.4) | 725 (31.6) | |

Note: SER—spherical equivalent of refraction; SD–standard deviation; D–diopters

**Table 2. The prevalence of different types of refractive errors by gender, school types and grades.**

| Demographic variable | Myopia, n (%) | p | Hyperopia, n (%) | p | Astigmatism, n (%) | p |
|---|---|---|---|---|---|---|
| Gender | | | | | | |
| Male | 319 (27.5) | 0.399 | 40 (3.4) | 0.815 | 33 (2.8) | 0.881 |
| Female | 329 (29.1) | | 37 (3.3) | | 31 (2.7) | |
| Grade | | | | | | |
| 1st | 135 (17.6) | <0.001 | 36 (4.7) | 0.044 | 17 (2.2)752 | 0.425 |
| 5th | 207 (27.0) | | 20 (2.6) | | 22 (2.9) | |
| 9th | 306 (40.5) | | 21 (2.8) | | 25 (3.3) | |
| School type | | | | | | |
| Gymnasiums | 309 (29.1) | 0.409 | 31 (2.9) | 0.279 | 29 (2.7) | 0.871 |
| General education schools | 339 (27.5) | | 46 (3.7) | | 35 (2.8) | |
| Total | 648 (28.3) | | 77 (3.4) | | 64 (2.8) | |

As seen from Table 3, there was an increase in the prevalence of all refractive errors and myopia in particular in older grades in both school types (p<0.05). We did not observe an association between hyperopia and astigmatism with the grade of education. SER among 9th graders of general education schools was -0.84±1.70D, significantly higher among 9th graders from gymnasiums and equaled -1.08±1.99D.

When different types of schools were compared, there were no significant differences in the prevalence of refractive errors (myopia, hyperopia and astigmatism) between the corresponding grades of gymnasiums and secondary schools (p>0.05 for all grades).

The odds of myopia were higher among the children in 5th grades (OR 1.73; 95% CI 1.36; 2.21) and 9th grade (OR 3.19; 95% CI 2.52; 4.04) as compared to first graders. We also observed dramatic increase of moderate myopia in 5th grade group (OR 2.26; 95% CI 1.22; 4.19) and in 9th grades group (OR 4.18; 95% CI 2.35; 7.44), when comparing with 1st grades children. The risk of high myopia significantly increased in the 9th grade students (OR 10.42; 95% CI 2.43; 44.73). Hyperopia among students in 5th grade (2.6%) was twice less prevalent compared to 1st grade (OR 0.54; 95% CI 0.31; 0.95).

Response rate to the questionnaires filled out by the parents on behavioral risk factors and the role of parental myopia was 50.9% (1166 subjects). As seen from Table 4, the prevalence of myopia among students whose parents were not myopic was 25.2%. Compared with having parents without myopia, having at least one parent with myopia was a risk factor for myopia (OR 1.50; 95% CI 1.08; 2.09) in the crude regression analysis, whereas the presence of myopia

**Table 3. The prevalence of refractive errors in students of different grades.**

| Grade | Mean SER ± SD, D | Emmetropia, n (%) | Refractive errors, n (%) | Myopia, n (%) | High myopia, n (%) | Hyperopia, n (%) | Astigmatism, n (%) |
|---|---|---|---|---|---|---|---|
| General education schools | | | | | | | |
| 1st | -0.11±1.11 | 316 (76.5) | 97 (23.5) | 72 (17.4) | 1 (1.4) | 25 (6.1) | 10 (2.4) |
| 5th | -0.40±1.26 | 288 (71.6) | 114 (28.4) | 105 (26.1)* | 1 (1.0) | 9 (2.2) | 9 (2.2) |
| 9th | -0.84±1.70 | 242 (58.2) | 174 (41.8) * | 162 (38.9) * | 7 (4.3)* | 12 (2.9) | 16 (3.8) |
| Gymnasium | | | | | | | |
| 1st | -0.27±1.01 | 282 (79.2) | 74 (20.8) | 63 (17.7) | 1 (1.6) | 11 (3.1) | 7 (2.0) |
| 5th | -0.57±1.60 | 253 (69.1) | 113 (30.9)* | 102 (27.9)* | 6 (5.9) | 11 (3.0) | 13 (3.6) |
| 9th | -1.08±1.99 | 187 (55.0) | 153 (45.0)* | 144 (42.4)* | 13 (9.0) | 9 (2.6) | 9 (2.6) |

Note: SER—spherical equivalent of refraction; SD–standard deviation; D–diopters. p-value was calculated for two identical grades of different types of schools:
*p <0.05

**Table 4. Risk factors of myopia from regression analysis.**

| Risk factor | Crude | | | Adjusted[a] | | |
|---|---|---|---|---|---|---|
| | OR | 95% CI | p | OR | 95% CI | p |
| **Gender** | | | | | | |
| Male | Ref | | | Ref | | |
| Female | 1.09 | 0.91–1.30 | 0.399 | 1.06 | 0.80–1.41 | 0.668 |
| **Grade** | | | | | | |
| 1st grade | Ref | | | Ref | | |
| 5th grade | 1.73 | 1.36–2.21 | <0.001 | 1.78 | 1.26–2.52 | <0.001 |
| 9th grade | 3.19 | 2.52–4.04 | <0.001 | 3.34 | 2.31–4.82 | <0.001 |
| **Parental myopia** | | | | | | |
| neither | Ref | | | Ref | | |
| either | 1.50 | 1.08–2.09 | 0.015 | 1.38 | 0.98–1.94 | 0.063 |
| both | 1.63 | 0.88–3.00 | 0.117 | Ref | | |
| **School type** | | | | | | |
| General education school | Ref | | | - | | |
| Gymnasium | 1.07 | 0.90–1.29 | 0.409 | - | | |
| **Uses smartphone (h per day)** | | | | | | |
| no | Ref | | | | | |
| <1 hour per day | 1.06 | 0.64–1.75 | 0.836 | - | | |
| 1.01–2.00 | 1.23 | 0.75–2.02 | 0.405 | - | | |
| > 2.01 | 1.6 | 0.95–2.67 | 0.074 | - | | |
| **Near work (h per day)** | | | | | | |
| <1 hour per day | Ref | | | Ref | | |
| 1.01–2.00 | 1.08 | 0.74–1.59 | 0.696 | Ref | | |
| > 2.01 | 1.54 | 1.05–2.24 | 0.026 | 1.16 | 0.87–1.55 | 0.317 |
| **Outdoors (h per day)** | | | | | | |
| <1 hour per day | Ref | | | Ref | | |
| 1.01–2.00 | 0.75 | 0.55–1.03 | 0.074 | Ref | | |
| > 2.01 | 0.61 | 0.43–0.86 | 0.005 | 0.64 | 0.46–0.89 | 0.009 |
| **School shift** | | | | | | |
| morning | Ref | | | Ref | | |
| afternoon | 1.36 | 0.98–1.89 | 0.068 | Ref | | |
| full day | 1.56 | 1.13–2.17 | 0.008 | 1.31 | 0.95–1.80 | 0.096 |
| **Sports activity** | | | | | | |
| no | Ref | | | Ref | | |
| yes | 0.71 | 0.54–0.94 | 0.016 | 0.70 | 0.52–0.93 | 0.015 |

Note: Ref—reference; OR–odds ratio; CI—confidence interval; h–hour; [a]adjusted models are adjusted for all the significant predictors in the crude models (school grade, myopia of one of the parents, near-work more than 2 hours a day, outdoors more than 2 hours a day, full school day and sports activity).

in both parents was not associated with the presence of myopia in a student (OR 1.63; 95% CI 0.88; 3.00).

Hours spent per day performing various activities in groups are shown in Table 4. Myopic children spent significantly more screen time, than did non-myopic children (OR 2.29; 95% CI 1.54; 3.42). Extracurricular near work over 2 hours per day increased the odds of myopia (OR 2.29; 95% CI 1.54; 3.42). The prevalence of myopia among students who spent more than 2 hours daily on outdoor activities was 22.2%, whereas that among students who spent less than 1 hours a day outdoors was 32.0%. Non-myopic children spent more total hours outdoors (OR 0.61; 95% CI 0.43; 0.86) than did myopic children (p<0.01). More sports activities were

also a protective factor (OR 0.65; 95% CI 0.49; 0.85). A number of selected predictors were associated with myopia in crude models (Table 4). In the multivariate adjusted regression analysis, significant positive predictors of myopia were the higher class-level: 5[th] grade (OR 1.78; 95% CI 1.26; 2.52) and 9[th] grade (OR 3.34; 95% CI 2.31; 4.82). Outdoor activity more than 2 hours a day (OR 0.64; 95% CI 0.46; 0.89) and sports activity (OR 0.70; 95% CI 0.52; 0.93) were protective against myopia in such multivariate adjusted analyses.

## Discussion

This study was designed to estimate the prevalence of refractive errors in school children in Almaty, Kazakhstan. The prevalence of refractive errors was 31.6%, and the most common refractive error was myopia, which accounted for 89.4% of all refractive errors.

In general, the prevalence of myopia among schoolchildren was 28.3% (95% CI 26.5; 30.1), indicative of the growing trend compared to the previous study in 2010 with 22.1% prevalence in the same age group [35]. We believe that constant increase in educational workloads, leading to more time spent for reading and studying with concurrent reduction in time spent outdoors can explain that [43–46]. Differences in examination methods or discrepancies in myopia definition and thresholds [39] can also explain inconsistent results of studies elsewhere. For example, refractive errors in previous studies were only determined by a spherical refraction component and equivalent refraction, but not SER. In addition, refraction was measured using retinoscopy as opposed to autorefraction [31–35]. Although both methods are reliable [47, 48], direct comparison between studies may not be easy. As expected, our results demonstrated an increase of the prevalence and severity of myopia throughout each successive grades: from 17,6% myopic and 0,3% high myopic students in the 1st grades to 40.5% and 2.5% in 9[th] grades, respectively. Moreover, 5[th] grade (OR 1.78, 95% CI 1.26; 2.52) and 9[th] grade (OR 3.34, 95% CI 2.32; 4.83) were significant factors for myopia progression in an adjusted analysis.

Furthermore, our findings were consistent with other studies in children confirming growing prevalence of myopic refraction and a decrease in the rate of hyperopia with age, especially in the second decade of life [8, 49–57]. Our study did not show the association of myopia with gender, as in some previous studies [58], but data were not consistent across other reports [59]. Such disagreements may indicate both cultural and behavioral characteristics of certain regions, for example, greater sports activity among boys or a more diligent attitude towards school among girls.

Another major change in lifestyle over the past 10 years has been the increase in extracurricular activities [60], decrease of the time spent on housework activities, outdoor activities and sports [61, 62]. This is confirmed by the significantly lower prevalence of myopia in rural areas and economically undeveloped areas, where time spent outside and for household chores is greater compared to the city [63–65]. However, recent research does not find a clear relationship between myopia and screen time [66].

This is the first study of myopia risk factors in schoolchildren in Kazakhstan, in which the effect of independent associated risk factors on myopia was assessed using multiple logistic regression. Contrary to the some available data on the association of myopia with near work range and using smartphones, there was no such relationship in our study [55]. Myopia prevalence, which increased with class-level (age), was almost 30% lower among children attending sports clubs and spending more than 2 hours a day outdoor. These results underscore the significance of environmental factors for myopia and confirm the need for a balance between workload and physical activity. Increased time outdoors are the interventions that are proven to reduce the onset of myopia and are simple interventions to implement in public schools and

at home [67]. Of note, outdoor activities for myopia prevention were implemented in several countries [68–72].

Our study did not confirm the association of myopia in a child with having both myopic parents. Indeed, the OR for both parents was higher than for one (1.63 versus 1.50), but it was statistically non-significant. However, a more likely explanation of such finding was the subjectivity when filling out the questionnaires, resulting in the exposure misclassification. Besides, there might have been a lack of parental awareness; some respondents could have mistaken for myopia any refractive errors requiring spectacle correction, such as presbyopia or hyperopia. In addition, it was likely that not all respondents knew the term 'myopia', instead using non-scientific 'nearsightedness' or 'short-sightedness', which could also affect the results of the study. Therefore, subsequent studies would need a greater sample size and provide an explanation in the myopia questionnaire for more accurate and reliable results.

The advantages of the study were large sample, strict adherence to the study protocol, including clear randomization, and determination of cycloplegic refraction. In general, the protocol of our study was close to RESC with some exceptions, including the use of optotypes in the form of Landolt C rings to determine visual acuity; the sample in a continuous array, and the inclusion of all students of the corresponding grades [38].

The study had several limitations. The study was conducted in governmental schools only. Because a number of studies have shown the association of socioeconomic status with myopia [73–75], private schools with children from families with higher socioeconomic status would likely have more myopia. Our study was cross-sectional and, therefore, we were unable to establish causal relationship between the associations we observed in the risk factor analyses. Moreover, our measurements were taken once at the beginning of the school year, in autumn. It may have biased the effect reducing myopia prevalence, because children may spend more time outdoors in the warm season, particularly during summer holidays with more exposure to natural daylight [76]. With regard to classification of near work in the questionnaire we used, we only asked about extracurricular part of it, which may entail some classification bias and thus is another limitation of our presentation. Furthermore, some selection bias may be present because we looked at the worst eye to diagnose disorders. Finally, we assessed only one urban region of Kazakhstan. The prevalence of myopia was likely higher in cities than in rural areas [77]. Thus, data from this study cannot be extrapolated to the rural or smaller cities of Kazakhstan, and further research is needed to ascertain the prevalence and risk factors in other regions. In addition, this study did not consider the association of myopia with ethnicity. Thus, to get a complete picture of the vision in schoolchildren in the country, research is needed in other regions of Kazakhstan.

## Conclusion

In summary, our study demonstrated that more than a third of school-age children in Almaty had refractive errors, whereas myopia was most common. Compared to previous local studies in this age group, myopia prevalence among urban children likely increased over the past 10 years. We also found that time spent outdoors may be associated with reduced odds of myopia; however, the effects we identified should be further studied in rural population of Kazakhstan.

## Acknowledgments

We would like to thank all survey staff of the Kazakh Eye Research Institute and Faculty of medicine and health care of al-Farabi Kazakh National University. We thank the education department of Almaty and schools' staff for their help in organizing the study. We are grateful to the reviewers and editor for their careful review and insightful comments.

## Author Contributions

**Conceptualization:** Ainagul Mukazhanova, Denis Vinnikov.

**Data curation:** Ainagul Mukazhanova, Neilya Aldasheva.

**Formal analysis:** Aliya Ualiyeva, Kaini Baigonova.

**Methodology:** Ainagul Mukazhanova, Neilya Aldasheva, Juldyz Iskakbayeva, Raushan Bakhytbek, Aliya Ualiyeva, Damet Ongarbaeva, Denis Vinnikov.

**Project administration:** Juldyz Iskakbayeva.

**Resources:** Kaini Baigonova.

**Software:** Kaini Baigonova.

**Supervision:** Ainagul Mukazhanova, Neilya Aldasheva, Raushan Bakhytbek, Damet Ongarbaeva.

**Validation:** Juldyz Iskakbayeva, Aliya Ualiyeva, Kaini Baigonova, Damet Ongarbaeva.

**Writing – original draft:** Ainagul Mukazhanova, Denis Vinnikov.

**Writing – review & editing:** Neilya Aldasheva, Juldyz Iskakbayeva, Raushan Bakhytbek, Aliya Ualiyeva, Kaini Baigonova, Damet Ongarbaeva.

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
