## [Decision Letter · Decision Letter 0]

20 Jan 2022

PONE-D-21-32174Prevalence of refractive errors and risk factors for myopia in schoolchildren of Almaty, Kazakhstan: a cross-sectional studyPLOS ONE

Dear Dr. Vinnikov,

Thank you for submitting your manuscript to PLOS ONE. After careful consideration, we feel that it has merit but does not fully meet PLOS ONE’s publication criteria as it currently stands. Therefore, we invite you to submit a revised version of the manuscript that addresses the points raised during the review process.

Both reviewers have provided detailed reviews.  Please consider their comments when forming your response, in particular the comments regarding discussion of previous literature.

**We look forward to receiving your revised manuscript.**

Kind regards,

Manbir Nagra

Academic Editor

PLOS ONE

Journal Requirements:

Reviewers' comments:

Reviewer's Responses to Questions

**Comments to the Author**

1. Is the manuscript technically sound, and do the data support the conclusions?

Reviewer #1: Yes

Reviewer #2: Partly

2. Has the statistical analysis been performed appropriately and rigorously? 

Reviewer #1: Yes

Reviewer #2: I Don't Know

3. Have the authors made all data underlying the findings in their manuscript fully available?

Reviewer #1: Yes

Reviewer #2: Yes

4. Is the manuscript presented in an intelligible fashion and written in standard English?

Reviewer #1: Yes

Reviewer #2: No

5. Review Comments to the Author

Reviewer #1: Title: Prevalence of refractive errors and risk factors for myopia in schoolchildren of Almaty, Kazakhstan: a cross-sectional study (PONE-D-21-32174)

Mukazhanova A, et al., presented a manuscript analyzing the myopia prevalence and risk factors in individuals aged 6-16 from the largest city of Kazakhstan. The authors employed both clinical measurements and questionnaire to collect data from 2293 individuals, and found that students’ grade, outdoor activity duration and sport are associated with myopia. This study provides valuable data for myopia research, and minor revision is needed.

Questions:

1. Line 43-45: mix data from different age groups to reach the conclusion of "Southeast Asia has been shown to have the greatest fraction of population with myopia and astigmatism whereas hyperopia in children ..." can be misleading, especially when this sentence is following the statement of Line 41-43. Please focus on the results of children studies.

2. L189: Typo, should be “Table 1” here?

3. L195: this paragraph is the description for Table 2, The prevalence of different types of refractive errors by gender, school types and grades

Reviewer #2: This is an interesting article looking at the prevalence and risk factors for myopia in a sample within Almaty, Kazakhstan. Generally speaking it is a good study, worthy of publication. However, there are some concerns and queries about what has been written and the article would benefit from edits and additional information. Some word choices and statements are not typical, and would benefit from review. A list of some of these is below:

Abstract

introduction: It appears that a study on refractive error prevalence has been published before? https://articlekz.com/en/article/24144 This statement needs adjusting as it is a rather bold claim.

Line 25 methods: What is a school shift? This wouldn’t be generally understood.

Line 33 Results: Large claim that time outdoors and sports are protective against myopia. None of the research in this paper can validate that claim, as it is just a cross-sectional observational study. I appreciate the context within other papers and the theory, but you cannot write this in the results and should be cautious of it again in your conclusion.

Introduction:

Line 41: myopia being the second most common cause of blindness is a worthy point, but given that it can be corrected with spectacles, should this not state that it’s reversible blindness or similar?

Line 44: Word choice: fraction may not be best

Line 49: Although it’s great that you’ve put percentages here to discuss the prevalence in Asia, you did not do this for other comments you’ve made here in the introduction on prevalence of refractive errors such as hyperopia in the Americas. Please remove or add the others percentages for consistency. Furthermore, the studies and prevalence/percentages stated are from studies that include subsets of population, which isn’t representative. For example the 96.5% from Korea is from young men enlisted, and many of these studies do not have rural area representation, which may bias. I’d be cautious here.

Line 51: reword: the greatest burden of refractive error is myopia.

Line 54: The statement of 19.7% of the population being highly myopic in 2050 is wrong from the source claimed. Please revisit and revise.

Line 60: these are really interesting points on the lack of publication and the differences noted in the former soviet union. Could reasons for this be discussed, is this due to ethnicity, or data collection methods?

Line 61: Thus doesn’t fit here as grammatically correct.

Line 66: phrasing such as ‘so far’ is informal. Generally this paragraph is written differently from the others with a colloquial phrasing, and needs changing.

Line 68: these are great prevalence data you’ve discussed, but don’t they show that the statement in the Abstract about this being the first paper to discuss refractive error is wrong!? Also, please expand on these papers, are they from particular age ranges etc? Furthermore, you’ve claimed that the results of these studies differ greatly; reading what you’ve written it’s only 0.6% different.. so please check again. I believe you need to rewrite these claims in the introduction as it appears there is data and evidence, but that the general information has key points missing and each study has limitations.

Methods

Line 84: These comments on the number of inhabitants and ages of children in ages etc. are good but need referencing for their sources.

Line 108: Great that you’ve clarified that the schools are state owned/ran. Please ensure that you expand on this limitation in your discussion as private schools with children from higher socioeconomic backgrounds would likely have more myopia due to the link of socioeconomic status and myopia.

Line 116: you can obtain data on orthok patients refractive error, but I believe you mean refractive error without correction. Please rephrase.

Line 123: Why only ask about extracurricular near work, and how have you ensured that this has been answered accurately? Isn’t this a limitation?

Line 121: Only in the discussion is the time of year for data collection mentioned. Many studies looking at time outdoors have had to adjust their questionnaires and methods etc. due to the seasonal effects of time outdoors. Please ensure you add what time of year the questionnaire was answered, as these will be needed to contextualise what the readers take from it.

Line 127: were the people conducting the data collection the same clinical team throughout?

Line 131: ‘specific background’. In what?

Line 132: There is a great amount of detail about the lighting in the classroom. With cyclo this is not as relevant, and therefore I’m unsure if this level of detail is necessary, given that the detail in other aspects of the methods is not visible.

Line 137: I believe they’re called Landolt C rings?

Line 141: pupil light reflex does not provide accommodation impact, would checking near vision have been better?

Line 145: I believe that the collection of data from the ‘worse eye’ is an extreme way to induce selection bias and potential over estimation of prevalence for refractive errors generally in the population. I would ask the editor and authors whether this process needs to change so that one eye was taken at random for the results using this data.

Line 151: You’ve mentioned the IMI protocol papers for defining the limit of myopia, which is understandable. However, there are different thresholds for hyperopia, with no reasoning or references given. Please could this be clarified. The same goes for astigmatism.

Line 164: Is the comment on the education department permission relevant, as I’m unsure if this is included in this study?

Line 171: was the Mann Whitney test used even if the data was normally distributed?

Line 175: I find it very surprising that there were no correlations between thing such as sports and time outdoors and near work and electronic use etc.

Results

Table 1. The P values for male and female myopia are exactly the same, is this correct?

Line 200: are these differences between schools etc significant? Please provide a P value

Line 209: The levels and differences between them (or lack of) are not shown, as no P values have been done and detail is limited. This also goes for the comment on the prevalence being higher in 9th grade, you need P values.

Table 3: There is a * in the gymnasium part of 5th grade high myopia, but this isn’t replicated anywhere else, and so it implies nothing is significant. This doesn’t appear right with the numbers that are listed (greater numbers in year 9 for example), and it’s likely that significance is due to lower numbers.

Line 219: what about high myopia, that also increased?

Line 224: A supplementary material of the questionnaire, or full verbatim statements of how the questions were raised would be important. As your results for myopic parents don’t match with other studies and expectation, it’s likely down to data collection error. Therefore, this should be included in the methods, and stated as a limitation further in the discussion.

Line 231: visual work is wrong, reword to near work. Also the tasks stated; these are not given earlier, how do the researchers know which tasks were included in the near work estimation? If these were measured directly or asked for, why have they not included this in more detail?

Line 235: statements for comparisons would benefit from P values

Line 242: please reformat the equations and detail here.

Table 4: the bands for low medium and high time outdoors and near work appear very rigid and with minimal distribution (2 hours and more than double the time 5 hours are in the same category for example). Where did the authors get their thresholds from?

Line 258: It is as expected, but likely down to age, this should be discussed too.

Line 278: the Local studies are discussed. Could the refractive thresholds used in these studies be mentioned?

Line 281: statement on threholds etc. throughout this paragraph need evidencing with references.

Line 284: Big claim that the study is important to develop a prevention plan. Issues with similar statements have been given before. All that the study is able to say is that myopia prevalence is higher than other refractive errors, and that this may be cause for concern. Not for developing a prevention plan.

Line 300: Please evidence where sport itself when not a substitute for time outdoors has been shown to protect against myopia. This is generally considered a tenuous link.

Line 305: limitation of questionnaires discussed, quotes and exerts as mentioned above are needed.

Line 313: What is RESK? Definition needed

Line 318: Here it now only apparent that 50% of participants responded for the questionnaire. This is not apparent at all from the study and methods and results, and therefore can be misleading. This needs to be clearer, and likely means that you only have a subset population that you’ve done analysis for risk factors on, not the full sample.

Line 328: Conclusion phrasing is atypical e.g. starting with thus, and claims of the serious health concern are bold and unfounded. This study did NOT demonstrate that it is a serious health problem, it just demonstrates prevalence and linked potential risk factors. Comments on why it causes health concerns or worries for healthcare are not discussed at all, and no protective factors are able to be measured. The whole section needs rewriting and a more conservative approach for what the paper has done and why it is useful.

6. PLOS authors have the option to publish the peer review history of their article (what does this mean?). If published, this will include your full peer review and any attached files.

Reviewer #1: No

Reviewer #2: No

---

## [Author Response · Author response to Decision Letter 0]

15 Feb 2022

PONE-D-21-32174

Prevalence of refractive errors and risk factors for myopia in schoolchildren of Almaty, Kazakhstan: a cross-sectional study

PLOS ONE

To the Editor:

We appreciate the reviewers’ responses and the Editor’s comments. In the following detailed response, we address each critique calling for changes point-by-point, indicating where relevant additional text has been added to the body of the manuscript and its location. We hope that this has improved the manuscript.

Reviewer #1: 

1. Questions:

Line 43-45: mix data from different age groups to reach the conclusion of "Southeast Asia has been shown to have the greatest fraction of population with myopia and astigmatism whereas hyperopia in children ..." can be misleading, especially when this sentence is following the statement of Line 41-43. Please focus on the results of children studies.

We thank you for your comment and agree that the sentence was misleading. Therefore, we have rephrased the sentence to read: ‘Childhood myopia takes the lead in some countries of Southeast Asia with prevalence reaching 80% among adolescents, whereas hyperopia in children may be most prevalent in the Americas’.

2. L189: Typo, should be “Table 1” here?

Yes, thank you for catching this up. Now corrected to "Table 1"

3. L195: this paragraph is the description for Table 2, The prevalence of different types of refractive errors by gender, school types and grades

Thanks for the comment. Indeed, that paragraph describes table 2, and have now amended the body text to be more consistent with the corresponding Table. Now this paragraph reads: “The overall prevalence of myopia was 28.3% (95% CI 26.5; 30.1). Of all participants with myopia, 79.2% had low myopia, 16.4% had moderate myopia, and 4.5% had high myopia. High myopia was more frequent among students of gymnasiums than among general education schools (6.5% vs 2.7%, p=0.014), while low myopia was more often observed in students at general education schools. Prevalence of hyperopia was 3.4%. Low hyperopia was found in 89.6% and moderate hyperopia was detected in 10.4% of all hyperopic participants. As demonstrated in Table 2, there were no significant differences in the prevalence of myopia and hyperopia among boys and girls.”

Reviewer #2: 

Abstract:

1. introduction: It appears that a study on refractive error prevalence has been published before? https://articlekz.com/en/article/24144 This statement needs adjusting as it is a rather bold claim.

Thank you for your important note. This article was also published by us, in which we showed the intermediate results of the research (pilot project), which included a limited number of children and schools. In addition to a greater number of respondents, our current study also examines risk factors for myopia. We also edited the introduction with the aim to better integrate preceding data from Kazakhstan.

2. Line 25 methods: What is a school shift? This wouldn’t be generally understood.

In Kazakhstan, as in other countries of the former USSR, public schools have a double-shift system. In a double-shift system, schools divide all covered population into two groups. Each group uses the same buildings, equipment and teachers, but at different times of the day. Pupils of the first shift usually attend school from early morning until mid-day, whereas the second group in in classes from mid-day to late afternoon. However, there also exist schools with full-day classes where children stay in class from morning to evening. In the present study, we compared two shifts with each other, because there is a difference in lighting, daylight hours and daily routine (time to wake up and go to bed). We have now provided more information on that in Methods within the body text, not in the abstract.

3. Line 33 Results: Large claim that time outdoors and sports are protective against myopia. None of the research in this paper can validate that claim, as it is just a cross-sectional observational study. I appreciate the context within other papers and the theory, but you cannot write this in the results and should be cautious of it again in your conclusion.

Thank you for your comment. We agree that the wording is not accurate. Changed to «were associated with a lower incidence of myopia»

4. Introduction:

Line 41: myopia being the second most common cause of blindness is a worthy point, but given that it can be corrected with spectacles, should this not state that it’s reversible blindness or similar?

Thanks for the important note! We rephrased this sentence, and now it reads: «Uncorrected refractive errors are the leading cause of moderate to severe visual impairment worldwide and the second most common cause of avoidable blindness»

5. Line 44: Word choice: fraction may not be best

Thank you for your comment. The sentence is revised and corrected: «Childhood myopia takes the lead in some countries of Southeast Asia with prevalence reaching 80% among adolescents, whereas hyperopia in children may be most prevalent in the Americas»

6. Line 49: Although it’s great that you’ve put percentages here to discuss the prevalence in Asia, you did not do this for other comments you’ve made here in the introduction on prevalence of refractive errors such as hyperopia in the Americas. Please remove or add the others percentages for consistency. Furthermore, the studies and prevalence/percentages stated are from studies that include subsets of population, which isn’t representative. For example the 96.5% from Korea is from young men enlisted, and many of these studies do not have rural area representation, which may bias. I’d be cautious here.

Thank you for your comment. The paragraph has been revised and corrected

7. Line 51: reword: the greatest burden of refractive error is myopia.

Thank you for your comment. The sentence has been corrected. 

8. Line 54: The statement of 19.7% of the population being highly myopic in 2050 is wrong from the source claimed. Please revisit and revise.

Thank you for your comment. We do apologize for the typo. Corrected by 9.8%

9. Line 60: these are really interesting points on the lack of publication and the differences noted in the former soviet union. Could reasons for this be discussed, is this due to ethnicity, or data collection methods?

This is indeed a multifactorial problem. It can be assumed that the differences in the data are related both to different geographical and climatic conditions, and to the ethnic and cultural characteristics of the peoples living in the countries of the former USSR. In addition, there are no uniform methods to collect and process data, even within the same country. To elaborate on that, we have now added a sentence in the end of the paragraph which reads: “Inconsistent data from the countries across the former Soviet Union may results from varying geographical and climatic conditions in addition to differences in ethnic and cultural composition of the peoples living in these countries”.

10. Line 61: Thus doesn’t fit here as grammatically correct.

We apologize for the mistake. Corrected to "For example"

11. Line 66: phrasing such as ‘so far’ is informal. Generally this paragraph is written differently from the others with a colloquial phrasing, and needs changing.

Thank you for your comment. We have now rephrased the sentence to read: “In the last 20 years, a number of studies reported the prevalence of visual impairments in school-age children in Kazakhstan”

12. Line 68: these are great prevalence data you’ve discussed, but don’t they show that the statement in the Abstract about this being the first paper to discuss refractive error is wrong!? Also, please expand on these papers, are they from particular age ranges etc? Furthermore, you’ve claimed that the results of these studies differ greatly; reading what you’ve written it’s only 0.6% different. so please check again. I believe you need to rewrite these claims in the introduction as it appears there is data and evidence, but that the general information has key points missing and each study has limitations.

Thanks for the comment. We have changed the abstract about the primacy of our study. Indeed, studies of visual impairment and refraction among children have been conducted before. Many of them were not published and were used only for internal statistics and reports in clinics. In addition, we have revised and corrected this paragraph to read: ’A cross-sectional study in the 2004 in Almaty, Kazakhstan revealed a 21% prevalence of refractive errors in schoolchildren, whereas myopia was confirmed in 14%, hyperopia in 3%, astigmatism in 1% and accommodation disorders in 3%. Another study (2010) reported that myopic refraction was found in 12% of rural school children and 22% among their urban counterparts. Recently, there were only a few studies of the prevalence and structure of refractive errors among schoolchildren in Kazakhstan.’

Methods

13. Line 84: These comments on the number of inhabitants and ages of children in ages etc. are good but need referencing for their sources.

We took the data from the website https://stat.gov.kz/official/industry/61/statistic/6 and from the official statistical collection (http://www.rcrz.kz/index.php/ru/statistika-zdravookhraneniya-2). We used data for 2019-2020 (at the time of the study). The references are now listed in the reference list as 36-37. 

14. Line 108: Great that you’ve clarified that the schools are state owned/ran. Please ensure that you expand on this limitation in your discussion as private schools with children from higher socioeconomic backgrounds would likely have more myopia due to the link of socioeconomic status and myopia.

We have included this in the discussion as follows: “The study was conducted in governmental schools only. Because a number of studies have shown the association of socioeconomic status with myopia [71–73], private schools with children from families with higher socioeconomic status would likely have more myopia.”.

15. Line 116: you can obtain data on orthok patients refractive error, but I believe you mean refractive error without correction. Please rephrase.

Thank you for your comment. The sentence revised and corrected to read: «Students under orthokeratology treatment were excluded because their uncorrected visual acuity and uncorrected refractive error could not be obtained.».

16. Line 123: Why only ask about extracurricular near work, and how have you ensured that this has been answered accurately? Isn’t this a limitation?

Thank you for your comment. This is a very important question, which we also studied when created the questionnaire. In the questionnaire, we listed the most common types of extracurricular near work (please see item 33 below). However, we agree that there may be certain limitations. We agree that this should be discussed as a limitation and have now added a sentence to the Limitations: “With regard to classification of near work in the questionnaire we used, we only asked about extracurricular part of it, which may entail some classification bias and thus is another limitation of our presentation”. 

Another limitation of our report with regard to the questionnaire we used is 

17. Line 121: Only in the discussion is the time of year for data collection mentioned. Many studies looking at time outdoors have had to adjust their questionnaires and methods etc. due to the seasonal effects of time outdoors. Please ensure you add what time of year the questionnaire was answered, as these will be needed to contextualise what the readers take from it.

Thank you for highlighting this issue. We agree that the time of the year plays role, which is also related to the climate and different levels of insolation. The study was conducted between September to December. We have added this refinement to the methods and in the discussion section (paragraph 4, sentence 1 of Methods and sentence 4 in the Limitations paragraph).

18. Line 127: were the people conducting the data collection the same clinical team throughout?

Yes, the examination of all children in all schools and classes was carried out by one team (ophthalmologists and nurses). A sentence on that was added to the Examination protocol paragraph.

19. Line 131: ‘specific background’. In what?

This refers to the training of personnel for optometric examination of children and careful data recording. In accordance with the recommendations of RESC (protocol and manual of procedures to assessment of the prevalence of visual impairment attributable to refractive error or other causes in school children), a short training was held for the entire team at the study commencement. We've added a clarification to the Methods (the first paragraph of Examination protocol).

20. Line 132: There is a great amount of detail about the lighting in the classroom. With cyclo this is not as relevant, and therefore I’m unsure if this level of detail is necessary, given that the detail in other aspects of the methods is not visible.

Thanks for the comment. The study was conducted in classrooms, we included a description of the lighting for a general understanding of the conditions in which children learn. We have removed this paragraph as we agree this may be not of great value.

21. Line 137: I believe they’re called Landolt C rings?

Yes, they can be called Landolt ring or Landolt broken ring or Landolt C. We have edited the text accordingly.

22. Line 141: pupil light reflex does not provide accommodation impact, would checking near vision have been better?

We aimed to say that we measured the size of the pupil with a school ruler to verify complete dilation and cycloplegia. Refraction examination protocol in children recommends refraction assessment with a pupil wider than 6 mm. In addition, cycloplegia was verified by no pupil response to light.

23. Line 145: I believe that the collection of data from the ‘worse eye’ is an extreme way to induce selection bias and potential over estimation of prevalence for refractive errors generally in the population. I would ask the editor and authors whether this process needs to change so that one eye was taken at random for the results using this data.

In our opinion, choosing the best eye, focusing on the right or left eye, as well as random selection, seriously entails high risk of missing disorders that can lead to other disease, including impaired binocular vision, amblyopia, strabismus and disability. To better proceed with this discussion, we have now reviewed the literature and conclude that choosing the worst eye is a common practice, especially when assessing visual impairment in children. Furthermore, in accordance with the recommendations of the RESC, "Individuals were considered myopic if one or both eyes were myopic, and hyperopic if one or both eyes were hyperopic, so long as no eye was myopic.". Accordingly, for a child to be considered myopic or hyperopic, having refractive error unilaterally is sufficient. In other words, we aimed to identify pupils even if the pathology was in one eye only. This will allow better accuracy in estimating the total number of visually impaired children. However, we agree that if there is visual impairment in both eyes, choosing the worst eye will shift the SER and increase the number of severe cases. Hence, that was the rationale to include the worst eye in the analysis, even if this can entail some selection bias. In response to the pertinent comment of the Reviewer, we have elected to add one more sentence to the limitations as follows: “Furthermore, some selection bias may be present because we looked at the worst eye to diagnose disorders”. 

24. Line 151: You’ve mentioned the IMI protocol papers for defining the limit of myopia, which is understandable. However, there are different thresholds for hyperopia, with no reasoning or references given. Please could this be clarified. The same goes for astigmatism.

This is indeed a very important notation because the thresholds for astigmatism and hyperopia vary widely across studies. We reviewed the studies and found that hyperopia was defined as a SER from + 0.5 to + 3.0D. Astigmatism in different studies was defined as cylindrical refraction from 0.5 to 1.5D. We chose to use the thresholds reported in the E3 - European Eye Epidemiology Consortium systematic review, which defined the threshold for hyperopia as SER ≥ 1.0D; and astigmatism threshold as cylindrical refraction ≥ 1.0 D in any meridian. 

https://www.ncbi.nlm.nih.gov/pmc/articles/PMC4385146/

We have added a reference to the study in the Methods (Definitions). 

25. Line 164: Is the comment on the education department permission relevant, as I’m unsure if this is included in this study?

The Department of Education in our country controls the work of schools. Permission from the city's Department of Education was required for the study to take place in schools, so we included it in the legal information section. Removed.

26. Line 171: was the Mann Whitney test used even if the data was normally distributed?

We are sorry that at some earlier stage of internal revision part of the text in statistical methods dropped out. Indeed, when data were normally distributed, we used t-test as an alternative for Mann-Whitney test. The sentence is now corrected to read: “Between-group comparisons of binary variables were tested using χ2 test from contingency tables, whereas continuous variables were tested using t-test or Mann-Whitney U-test depending on the distribution”. 

27. Line 175: I find it very surprising that there were no correlations between thing such as sports and time outdoors and near work and electronic use etc.

This is indeed important, and we tried to consider multicollinearity in the models (outdoor time and sports activities; near work and full school day; near work and use of gadgets). Before we included the predictors in the adjusted model, we tested correlations using a correlation matrix. We could not use variance inflation factor or other tests of collinearity because of the binary nature of these variables. None of the correlations between variables included in the adjusted models exceeded 0.10; therefore, we failed to detect collinearity between predictors included.

28. Results

Table 1. The P values for male and female myopia are exactly the same, is this correct?

That is an unfortunate typo. Thank you for your comment. Corrected.

29. Line 200: are these differences between schools etc significant? Please provide a P value.

Thank you for your comment. The differences were significant (p = 0.014), and p-value is now provided.

30. Line 209: The levels and differences between them (or lack of) are not shown, as no P values have been done and detail is limited. This also goes for the comment on the prevalence being higher in 9th grade, you need P values.

Thank you for your comment. We apologize for this omission. Changed the table and this paragraph, which now reads: “As can be seen from Table 3, there was an increase in the prevalence of all refractive errors and myopia in particular in older grades in both school types (p < 0.05). We did not observe an association between hyperopia and astigmatism with the grade of education”.

31. Table 3: There is a * in the gymnasium part of 5th grade high myopia, but this isn’t replicated anywhere else, and so it implies nothing is significant. This doesn’t appear right with the numbers that are listed (greater numbers in year 9 for example), and it’s likely that significance is due to lower numbers.

We are grateful for your important remark. In Table 3, we calculated the significance for the same grades but different types of schools. Indeed, significant differences were found only for high myopia among grade 5. However, we agree that this may be due to the lower number of children with high myopia. We added p-value to the table for clarity and a text description was added to the Results: When different types of schools were compared, there were no significant differences in the prevalence of refractive errors (myopia, hyperopia and astigmatism) between the corresponding grades of gymnasiums and secondary schools (p>0.05 for all grades). 

 Mean SER ± SD, D Emmetropia, n (%) Refractive errors, n (%) Myopia, n (%) High myopia, n (%) Hyperopia, n (%) Astigmatism, n (%)

General education schools

1st -0.11±1.11 316 (76.5) 97 (23.5) 72 (17.4) 1 (1.4) 25 (6.1) 10 (2.4)

5th -0.40±1.26 288 (71.6) 114 (28.4) 105 (26.1) 1 (1.0) 9 (2.2) 9 (2.2)

9th -0.84±1.70 242 (58.2) 174 (41.8) 162 (38.9) 7 (4.3) 12 (2.9) 16 (3.8)

Gymnasium

1st -0.27±1.01 282 (79.2) 74 (20.8)

p = 0.370 63 (17.7)

p = 0.924 1 (1.6)

p = 0.917 11 (3.1)

p = 0.053 7 (2.0)

p = 0.669

5th -0.57±1.60 253 (69.1) 113 (30.9)

p = 0.446 102 (27.9)

p = 0.586 6 (5.9)*

p = 0.043 11 (3.0)

p = 0.506 13 (3.6)

p = 0.276

9th -1.08±1.99 187 (55.0) 153 (45.0)

p = 0.382 144 (42.4)

p = 0.342 13 (9.0)

p = 0.069 9 (2.6)

p = 0.844 9 (2.6)

p = 0.360

In the manuscript, we have replaced Table 3 with a comparison of the prevalence of refractive errors among different grades of education.

32. Line 219: what about high myopia, that also increased?

The risk of high myopia significantly increased in the 9th grade students (OR 10.42; 95% CI 2.43; 44.73). However, due to the limited number of cases of high myopia, 95% CI was so wide. We have added this point to the manuscript.

33. Line 224: A supplementary material of the questionnaire, or full verbatim statements of how the questions were raised would be important. As your results for myopic parents don’t match with other studies and expectation, it’s likely down to data collection error. Therefore, this should be included in the methods, and stated as a limitation further in the discussion.

Thank you, we agree that this information should have been added. The following information from the original questionnaire is now disclosed in the Methods:

- “Does any parent have myopia?” with three response options (no; one of the parents; 'both parents). We did not specify whether that was a father or mother or even familial myopia.

- “'How many hours does the child spend in the outdoor activities daily? “with three answer options (less than 1 hour a day, 1-2 hours a day, more than 2 hours a day). 

- “How many hours does the child spend on the near work (reading, drawing, handicraft, homework, etc.); time spent at school is not considered?” with three answer options (less than 1 hour a day, 1-2 hours a day, more than 2 hours).

- “How many hours does the child spend with gadgets (computers, mobile phones, tabs, games, etc.), daily’, with four variants of answers (doesn’t use gadgets, less than 1 hour a day, 1-2 hours a day, more than 2 hours).

- “Does your child attend sports club or section”, with two options to answer (yes, no)

- “What school shift is your child in?” with two answer options (shift one, shift two or whole day.

In addition, we included gender and school grade. But due to the word limit in a manuscript, we decided to add the following sentence to the abstract: “The questionnaire included questions on the main risk factors such as parental myopia, screen time, time outdoors, sports activities, near-work, gender, grade and school shift”.

34. Line 231: visual work is wrong, reword to near work. Also the tasks stated; these are not given earlier, how do the researchers know which tasks were included in the near work estimation? If these were measured directly or asked for, why have they not included this in more detail?

Thank you for your comment. The sentence is revised and corrected to read: “Extracurricular near work over 2 hours per day increased the odds of myopia (OR 2.29; 95% CI 1.54; 3.42)”. In addition, we now describe the tasks of near work in the Methods, as stated above.

35. Line 235: statements for comparisons would benefit from P values.

Thank you for your comment. We added "p<0.01" to the sentence.

36. Line 242: please reformat the equations and detail here.

We have to admit that initially we were not intended to provide any formula to calculate the odds of visual impairment and make it a function of mathematics. We were advised to put this formula by one of the internal reviewers whom we showed the manuscript in search of advice. We personally consider citing this formula impertinent, and the Reviewer’s comment only strengthened this concern. Therefore, at this stage we have decided to completely delete the formula from the text as it looks misleading and overloads the reader with information he or she is not in need of. 

37. Table 4: the bands for low medium and high time outdoors and near work appear very rigid and with minimal distribution (2 hours and more than double the time 5 hours are in the same category for example). Where did the authors get their thresholds from?

We agree with the Reviewer. However, when parents themselves fill the questionnaires in, they feel such clear time frames are more convenient for them to think of. When we made decision on what periods to use, we relied on data from previous studies, which also give hard time periods, including The North India Myopia Study (https://www.ncbi.nlm.nih.gov/pmc/articles/PMC4342249/); German KiGGS Study (https://www.ncbi.nlm.nih.gov/pmc/articles/PMC8025934/); Study by Stepan Rusnak et al. (https://www.ncbi.nlm.nih.gov/pmc/articles/PMC5859838/); Xiaoyan Wu et al. (https://www.ncbi.nlm.nih.gov/pmc/articles/PMC5006500/). 

38. Line 258: It is as expected, but likely down to age, this should be discussed too.

Thank you for your comment. We have expanded the discussion of myopia with age in the Discussion section. This insertion reads: “Furthermore, our findings were consistent with other studies in children confirming growing prevalence of myopic refraction and a decrease in the rate of hyperopia with age, especially in the second decade of life [8,47–55]”.

39. Line 278: the Local studies are discussed. Could the refractive thresholds used in these studies be mentioned?

Thanks for the clarification. Differences in thresholds were mainly for hypermetropia, which was defined as more then SER + 1.25 D. In addition, there were variations in methodology (retinoscopy vs autorefraction) and the definition of refractive errors only by the spherical component. We have added these points to the Discussion: “For example, refractive errors in previous studies were only determined by a spherical refraction component and equivalent refraction, but not SER. In addition, refraction was measured using retinoscopy as opposed to autorefraction [31–35]. Although both methods are reliable [45,46], direct comparison between studies may not be easy”.

40. Line 281: statement on threholds etc. throughout this paragraph need evidencing with references.

Thank you for your comment. Done. Please see item above citing the text inserted. 

41. Line 284: Big claim that the study is important to develop a prevention plan. Issues with similar statements have been given before. All that the study is able to say is that myopia prevalence is higher than other refractive errors, and that this may be cause for concern. Not for developing a prevention plan.

Thank you. This phrase has been removed.

42. Line 300: Please evidence where sport itself when not a substitute for time outdoors has been shown to protect against myopia. This is generally considered a tenuous link.

We thank the Reviewer for catching this up and addressing this important aspect. We have to admit that we did not distinguish between outdoor activities and sports, and only asked about outdoor activities. Hence, referring to sports is inappropriate and should be avoided. We have decided to delete the term ‘sport’ and retain ‘time spent outdoors’. 

43. Line 305: limitation of questionnaires discussed, quotes and exerts as mentioned above are needed.

Thank you. We have added a paragraph about the limitations of our study

44. Line 313: What is RESK? Definition needed

RESC definition is provided in the first sentence of the third paragraph in Methods as follows: “This cross-sectional school-based study was performed according to the protocol of Refractive Error Study in Children (RESC). This protocol was designed to standardize the methodology used to obtain prevalence data on childhood refractive errors”.

45. Line 318: Here it now only apparent that 50% of participants responded for the questionnaire. This is not apparent at all from the study and methods and results, and therefore can be misleading. This needs to be clearer, and likely means that you only have a subset population that you’ve done analysis for risk factors on, not the full sample.

Thanks for the comment. The low response rate is also a limitation of this study. We apologize for not informing the reader about 50% response rate earlier in the manuscript. And we have now decided to put this data in the Results: “Response rate to the questionnaires filled out by the parents on behavioral risk factors and the role of parental myopia was 50.9% (1166 subjects)”.

46. Line 328: Conclusion phrasing is atypical e.g. starting with thus, and claims of the serious health concern are bold and unfounded. This study did NOT demonstrate that it is a serious health problem, it just demonstrates prevalence and linked potential risk factors. Comments on why it causes health concerns or worries for healthcare are not discussed at all, and no protective factors are able to be measured. The whole section needs rewriting and a more conservative approach for what the paper has done and why it is useful.

Thank you for your comment. We agree with this remark and amended the paragraph to make out conclusions not sound too strong and present only those directly arising from the study findings. Now this paragraph reads: “In summary, our study demonstrated that more than a third of school-age children in Almaty had refractive errors, whereas myopia was most common. Compared to previous local studies in this age group, myopia prevalence among urban children likely increased over the past 10 years. We also found that time spent outdoors may be associated with reduced odds of myopia; however, the effects we identified should be further studied in rural population of Kazakhstan.”.

We are grateful to the editor and reviewers for their time and all the valuable comments, as well as for the help in improving our manuscript. We will make every effort to meet the requirements of the Journal. 

Corresponding author, Denis Vinnikov

---

## [Decision Letter · Decision Letter 1]

11 Apr 2022

PONE-D-21-32174R1Prevalence of refractive errors and risk factors for myopia among schoolchildren of Almaty, Kazakhstan: a cross-sectional studyPLOS ONE

Dear Dr. Vinnikov,

Thank you for submitting your manuscript to PLOS ONE. After careful consideration, we feel that it has merit but does not fully meet PLOS ONE’s publication criteria as it currently stands. Therefore, we invite you to submit a revised version of the manuscript that addresses the points raised during the review process.

ACADEMIC EDITOR: Please pay particular attention to the point regarding eye selection for statistical analysis. If sufficient evidence to support the approach used cannot be provided, then please consider reanalysis of data using approaches more aligned with established norms.

We look forward to receiving your revised manuscript.

Kind regards,

Manbir Nagra

Academic Editor

PLOS ONE

Reviewers' comments:

Reviewer's Responses to Questions

**Comments to the Author**

1. If the authors have adequately addressed your comments raised in a previous round of review and you feel that this manuscript is now acceptable for publication, you may indicate that here to bypass the “Comments to the Author” section, enter your conflict of interest statement in the “Confidential to Editor” section, and submit your "Accept" recommendation.

Reviewer #1: All comments have been addressed

Reviewer #2: (No Response)

2. Is the manuscript technically sound, and do the data support the conclusions?

Reviewer #1: Yes

Reviewer #2: Partly

3. Has the statistical analysis been performed appropriately and rigorously? 

Reviewer #1: Yes

Reviewer #2: Yes

4. Have the authors made all data underlying the findings in their manuscript fully available?

Reviewer #1: Yes

Reviewer #2: Yes

5. Is the manuscript presented in an intelligible fashion and written in standard English?

Reviewer #1: Yes

Reviewer #2: Yes

6. Review Comments to the Author

Reviewer #1: (No Response)

Reviewer #2: Thank you very much to the authors for their responses and changes to the manuscript. The manuscript is much improved, and I can now follow the process of the study and what has been performed. I believe it only requires minor revisions/clarifications for publication, and I would appreciate if the authors would be able to respond to the following points below.

1.Comment number 16 talking about limitations to the questionnaire, it appears that your response was truncated, when commenting on another limitation. Please could the authors state whether there is something else to comment on here, as these words cannot be seen after the new section written in the manuscript.

2.I believe that having understood the questions asked within the questionnaire, that there is a new limitation to add, unfortunately. Many children will not know what myopia is (indeed, most of the adult population may not either), and may usually know some form of non-scientific term such as ‘short-sightedness’ or similar. Given this wording to identify myopic parents, it may be that children did not understand the question, and this may be why the result on number of myopic parents may be not as expected. This should therefore be added to the discussion in this relevant part, as a limitation and a reason for the results found.

3.I appreciate the authors answer to point 22 on cycloplegia. No response to light is indeed one of the signs for potential cycloplegia. However, usually protocols for cyclo-autorefraction indicate a wait time also after the instillation of drops (sometimes more than one), for the reason that just stopping pupillary reactions is not enough to demonstrate there is no accommodation. Could the authors clarify if there was a wait time, or just if pupil reaction and size was only checked.

4.For the response to point 23, I appreciate the response from the authors to answer the fact that they have used the worse eye, and added a point to the limitations, however for myopia-related studies, this is not typical, and may be seen as strange. Many researchers would firmly state that this would inflate all values found, and ask why the authors didn’t do an analysis of just one eye, and believe that refractions/eye condition prevalence would level out due to laterality chance. Indeed, the authors have said they’ve reviewed literature that showed this was best approach. To placate the anticipated readership, please could they add this to their methods as explanation for their approach, along with references they’ve found to back this approach.

In conclusion, I understand where the authors are coming from, yet I still have reservations. I leave this up to the editor to decide what to do.

5.Point 27, thank you for clarifying. However, are you sure that the variables are binary? Do you mean categorical? And if so, did you only categorise in a yes/no binary format? Apologies if I have misunderstood this.

6.Point 28. Thanks for clarifying. Unfortunately the new manuscript demonstrates you’ve removed the female p value, but I cannot see the correct one.

7. PLOS authors have the option to publish the peer review history of their article (what does this mean?). If published, this will include your full peer review and any attached files.

Reviewer #1: No

Reviewer #2: No

---

## [Author Response · Author response to Decision Letter 1]

18 Apr 2022

PONE-D-21-32174R1

Prevalence of refractive errors and risk factors for myopia in schoolchildren of Almaty, Kazakhstan: a cross-sectional study

PLOS ONE

To the Editor:

We would like to thank the reviewers and the Editor for comments and help in improving our manuscript. In the following detailed response, we address each critique calling for changes point-by-point, indicating where relevant additional text has been added to the body of the manuscript and its location. We hope that this has improved the manuscript.

ACADEMIC EDITOR: Please pay particular attention to the point regarding eye selection for statistical analysis. If sufficient evidence to support the approach used cannot be provided, then please consider reanalysis of data using approaches more aligned with established norms.

Thank you for your comment. In accordance with the recommendations of the WHO and RESC, "Individuals were considered myopic if one or both eyes were myopic, and hyperopic if one or both eyes were hyperopic, so long as no eye was myopic.". Accordingly, for a child to be considered myopic or hyperopic, having refractive error unilaterally is sufficient. In our opinion, choosing the best eye, focusing on the right or left eye, as well as random selection, seriously entails high risk of missing disorders that can lead to other disease, including impaired binocular vision, amblyopia, strabismus and disability. To better proceed with this discussion, we have now reviewed the literature and conclude that choosing the worst eye is a common practice, especially when assessing visual impairment in children. For example, in 2006, the population multiethnic study of pediatric eye diseases (MEPEDS) was introduced, in which the determination of visual acuity and refraction was carried out in the worst eye (https://pubmed.ncbi.nlm.nih.gov/16877284/ ).

In many large studies, the selection of the worst eye is stated explicitly. In other studies, in order to be considered as a subject with refractive error, the participant had to have a refractive error in at least one eye. In cases with one myopic and a fellow hyperopic eye, the refractive error of the eye with larger absolute SE was taken into account. Which also means choosing the worst eye. These are just some of them:

https://www.ncbi.nlm.nih.gov/pmc/articles/PMC6941318/

https://www.ncbi.nlm.nih.gov/pmc/articles/PMC6029024/

https://pubmed.ncbi.nlm.nih.gov/25626973/

https://www.ncbi.nlm.nih.gov/pmc/articles/PMC5468969/

https://www.ncbi.nlm.nih.gov/pmc/articles/PMC8211073/

https://www.ncbi.nlm.nih.gov/pmc/articles/PMC8139091/

https://pubmed.ncbi.nlm.nih.gov/12395920/

https://www.ncbi.nlm.nih.gov/pmc/articles/PMC7197133/

https://www.ncbi.nlm.nih.gov/pmc/articles/PMC2815146/

https://www.ncbi.nlm.nih.gov/pmc/articles/PMC5468969/#CR22

https://www.ncbi.nlm.nih.gov/pmc/articles/PMC3853776/

https://journals.plos.org/plosone/article?id=10.1371/journal.pone.0120764

https://www.ncbi.nlm.nih.gov/pmc/articles/PMC8453643/

Moreover, in none of the above studies, the choice of the worst eye was a limitation.

All of the above explains the rationale for choosing the worst eye for examination. However, we agree that if there is visual impairment in both eyes, choosing the worst eye will shift the SER and increase the number of severe cases. In response to the pertinent comment of the Reviewer, we added the following paragraph to the Materials and Methods section: "Because the correlation of spherical equivalent and visual acuity between the eyes was high (Pearson correlation coefficient 0.91 (95% CI 0.90; 0.92; p<0.001)), the spherical equivalent from the worse eye was used for analysis, similar to several previous cross-section studies in children."

Reviewer #2: 

1. Comment number 16 talking about limitations to the questionnaire, it appears that your response was truncated, when commenting on another limitation. Please could the authors state whether there is something else to comment on here, as these words cannot be seen after the new section written in the manuscript.

Thank you for your comment. We acknowledge that we made a misprint in point 16. The phrase “Another limitation of our report with regard to the questionnaire we used is…” is the beginning of an answer to another question that erroneously fell into this paragraph. We apologize for the annoying misprint. The answer should have been: «Thank you for your comment. This is a very important question, which we also studied when created the questionnaire. In the questionnaire, we listed the most common types of extracurricular near work (please see item 33 below). However, we agree that there may be certain limitations. We agree that this should be discussed as a limitation and have now added a sentence to the Limitations: “With regard to classification of near work in the questionnaire we used, we only asked about extracurricular part of it, which may entail some classification bias and thus is another limitation of our presentation”. 

We have made these changes to the manuscript (line 317)

2. I believe that having understood the questions asked within the questionnaire, that there is a new limitation to add, unfortunately. Many children will not know what myopia is (indeed, most of the adult population may not either), and may usually know some form of non-scientific term such as ‘short-sightedness’ or similar. Given this wording to identify myopic parents, it may be that children did not understand the question, and this may be why the result on number of myopic parents may be not as expected. This should therefore be added to the discussion in this relevant part, as a limitation and a reason for the results found.

Thank you for your comment. We completely agree with you on this remark. We wrote about this in the discussion section: "Besides, there might have been a lack of parental awareness; some respondents could have mistaken for myopia any refractive errors requiring spectacle correction, such as presbyopia or hyperopia". We now decided to add a clarification on the terminology as follows: "In addition, it was likely that not all respondents knew the term 'myopia', instead using non-scientific 'nearsightedness' or 'short-sightedness', which could also affect the results of the study"

3. I appreciate the authors answer to point 22 on cycloplegia. No response to light is indeed one of the signs for potential cycloplegia. However, usually protocols for cyclo-autorefraction indicate a wait time also after the instillation of drops (sometimes more than one), for the reason that just stopping pupillary reactions is not enough to demonstrate there is no accommodation. Could the authors clarify if there was a wait time, or just if pupil reaction and size was only checked.

Thank you for the clarification, we probably misunderstood the question. We agree that exposure time is important for cycloplegia. In our study, the wait time after the instillation of drops was 30 minutes. We have described this in the 'Materials and Methods' section, paragraph 'Examination protocol': "We measured cycloplegic refraction 30 minutes after the instillation of a drop of cyclopentolate 1% twice."

4. For the response to point 23, I appreciate the response from the authors to answer the fact that they have used the worse eye, and added a point to the limitations, however for myopia-related studies, this is not typical, and may be seen as strange. Many researchers would firmly state that this would inflate all values found and ask why the authors didn’t do an analysis of just one eye, and believe that refractions/eye condition prevalence would level out due to laterality chance. Indeed, the authors have said they’ve reviewed literature that showed this was best approach. To placate the anticipated readership, please could they add this to their methods as explanation for their approach, along with references they’ve found to back this approach. In conclusion, I understand where the authors are coming from, yet I still have reservations. I leave this up to the editor to decide what to do.

We are grateful to you for your attention to this issue. We agree that this is a significant limitation of our study, and we must explain the choice of the worst eye for analysis. Please see clarification and rationale provided in response to the Editor’s comment above. To do this, we added the following paragraph to the Materials and Methods section: "Because the correlation of spherical equivalent and visual acuity between the eyes was high (Pearson correlation coefficient 0.91 (95% CI 0.90; 0.92; p<0.001)), the spherical equivalent from the worse eye was used for analysis, similar to several previous cross-section studies in children"

5. Point 27, thank you for clarifying. However, are you sure that the variables are binary? Do you mean categorical? And if so, did you only categorise in a yes/no binary format? Apologies if I have misunderstood this.

We are sorry for using improper term. Yes, we meant categorical variables and we have now corrected ‘binary’ to ‘categorical’. Some of these were categorized into ‘yes/no’, but other into categories, depending on the variable. 

6. Point 28. Thanks for clarifying. Unfortunately the new manuscript demonstrates you’ve removed the female p value, but I cannot see the correct one. 

We are truly sorry for choosing improper format for data presentation in Table 1. We have now edited the Table to make it self-explanatory. 

We have also noticed that the format should also be amended in Table 2 and have done that to be consistent with Table 1. 

We are grateful to the editor and reviewers for all further valuable comments, as well as for the help in improving our manuscript. We will make every effort to meet the requirements of the Journal. 

Corresponding author, Denis Vinnikov

---

## [Editor Report · Decision Letter 2]

23 May 2022

Prevalence of refractive errors and risk factors for myopia among schoolchildren of Almaty, Kazakhstan: a cross-sectional study

PONE-D-21-32174R2

Dear Dr. Vinnikov,

We’re pleased to inform you that your manuscript has been judged scientifically suitable for publication and will be formally accepted for publication once it meets all outstanding technical requirements.

Kind regards,

Manbir Nagra

Academic Editor

PLOS ONE
---

## [Editor Report · Acceptance letter]

24 May 2022

PONE-D-21-32174R2 

Prevalence of refractive errors and risk factors for myopia among schoolchildren of Almaty, Kazakhstan: a cross-sectional study 

Dear Dr. Vinnikov:

I'm pleased to inform you that your manuscript has been deemed suitable for publication in PLOS ONE. Congratulations! Your manuscript is now with our production department. 

Kind regards, 

on behalf of

Dr. Manbir Nagra 

Academic Editor

PLOS ONE